# Responsiveness of the Calf-Raise Senior test in community-dwelling older adults undergoing an exercise intervention program

Helô-Isa André[1☯¤a]*, Vera Moniz-Pereira[1¤a‡], Fátima Ramalho[2¤b‡], Rita Santos-Rocha[2¤b], António Veloso[1¤a], Filomena Carnide[1☯¤a]

**1** CIPER, Sports & Health Department, Biomechanics and Functional Morphology Laboratory, Faculty of Human Kinetics, University of Lisbon, Cruz Quebrada-Dafundo, Lisbon, Portugal, **2** Sport Sciences School of Rio Maior, Polytechnic Institute of Santarém, Rio Maior, Portugal

☯ These authors contributed equally to this work.
¤a Current address: Estrada da Costa, Cruz Quebrada-Dafundo, Lisbon, Portugal
¤b Current address: Avenida Dr. Mário Soares, Rio Maior, Portugal
‡ These authors also contributed equally to this work.
* heloandre@fmh.ulisboa.pt

**Data Availability Statement:** Additional data files are available from the Figshare database (https://figshare.com/s/2903587ad9d87f3d0d37).

## Abstract

### Introduction

Mobility significantly depends on the ankle muscles' strength which is particularly relevant for the performance of daily activities. Few tools are available, to assess ankle strength with all of the measurement properties tested. The purpose of this study is to test the responsiveness of Calf-Raise Senior Test (CRS) in a sample of elderly participants undergoing a 24-week community exercise program.

### Methods

82 older adults participated in an exercise program and were assessed with CRS Test and 30-second chair stand test (CS) at baseline and at follow-up. Effect size (ES), standardized response mean (SRM) and minimal detectable change (MDC) measures were calculated for the CRS and CS tests scores. ROC curves analysis was used to define a cut-off representing the minimally important difference of Calf-Raise Senior test.

### Results

Results revealed a small (ES = 0.42) to moderate (SRM = 0.51) responsiveness in plantar-flexion strength and power across time, which was lower than that of CS test (ES = 0.64, SRM = 0.67). The responsiveness of CRS test was more evident in groups of subjects with lower initial scores. A minimal important difference (MID) of 3.5 repetitions and a minimal detectable change (MDC) of 4.6 was found for the CRS.

### Conclusion

Calf-Raise Senior Test is a useful field test to assess elderly ankle function, with moderate responsiveness properties. The cutoff scores of MDC and MID presented in this study can

**Funding:** Helô-Isa André was supported by the Portuguese Foundation for Science and Technology (FCT) through the award of the doctoral grant SFRH/BD/62429/2009 (https://www.fct.pt/apoios/bolsas/index.phtml.en). Helô-Isa André; Vera Moniz-Pereira; Rita Santos-Rocha; António Veloso; Filomena Carnide were supported by the Portuguese Foundation for Science and Technology (FCT) through the award PTDC/DES/72946//2006 (https://www.fct.pt/apoios/projectos/consulta/areas.phtml.en?idElemConcurso=914) The funders had no role in study design, data collection and analysis, decision to publish, or preparation of the manuscript.

**Competing interests:** NO authors have competing interests

be useful in determining the success of interventions aiming at improving mobility in senior participants.

## Introduction

Population aging is a major global demographic trend. The increase in life expectancy raises the related concern of increasing morbidity, prolonged disability and dependency, with a reduction in quality of life [1]. This demographic transition is influencing the economy, care, social development, welfare and well-being. Thus, it is crucial to adapt policy, health services and intervention programs to an aging and frailer population.

The quality of life of the elderly depends on their health and ability to perform activities of daily living (ADLs) [2]. Mobility is a prerequisite for the performance of most common ADLs and its maintenance is a major goal of geriatric health professionals [3]. Mobility greatly depends on the strength in the lower limbs, especially in the ankle muscles, which are particularly relevant in gait function [2, 4, 5] climbing stairs and rising from a chair [5–7]. Plantar-flexors (PF) muscles act to support and to propel the body forward in late stance and their strength is positively related to gait velocity and step length [8]. Lower strength levels are very common in older adults [9, 10] and are associated with poor gait and balance [11, 12]. PF muscles, in particular, reveal large decline in strength with aging, presenting a decrease of 2.3% in very old adults ($>$85 years-old) per year [13], and differences of about 40% when comparing elderly with young men [14, 15].

PF assessment is especially important to design programs or implement strategies, aiming the preservation of the mobility. This issue is even more relevant in the design of exercise programs tailored to functional status of older adults [16–18]. In order to ensure meaningful and quality data related to the functional loss and changes in mobility parameters with age, the use of measurement tools with relevant psychometric properties is essential. Studies reporting validity and responsiveness of strength and mobility assessment tools are relatively scarce [19, 20]. Additionally, few of the aforementioned studies report sensibility and sensitivity data, allowing the establishment of the minimal importance difference. To our knowledge, only the "30 second chair stand test" (CS) [19] and the "Timed Up and Go" (TUG) [20] have been tested for responsiveness in community-setting interventions with healthy and independent older adults. Although both tests are used to assess lower limbs muscle function, none of them provides specific information about the PF strength, which has already been mentioned as being relevant for preserving the quality of gait in the elderly [2, 4–5]

The Calf-Raise Senior (CRS) test is the only field assessment tool developed to evaluate ankle muscle function in the elderly and has shown to have an excellent test-retest reliability (ICC = 0.90), inter-rater reliability (ICC = 0.93–0.96) and a good intra-rater reliability (ICC = 0.79–0.84) [21,22]. The test also presented a significant association between its scores and laboratory strength assessments (isometric, r = 0.87, $r^2$ = 0.75; isokinetic, r = 0.86, $r^2$ = 0.74; and rate of force development, r = 0.77, $r^2$ = 0.59), demonstrating to be an excellent indicator of ankle strength in older adults [21]. Despite good results regarding the reliability and validity of the CRS test [21,22], its responsiveness has not yet been established. Therefore, the purpose of this study is to test the responsiveness of the CRS test in a sample of elderly participants undergoing a 24 weeks' community exercise program.

## Materials and methods

### Study design and subjects

A prospective multi-site cohort study was conducted, with a 24-week follow-up period. The follow up interval was defined by having into account that although major adaptations in strength and power in the elderly occur after 12 weeks of training [23–25] higher effect has been showed after longer training periods (≥24 weeks) [26]. All participants underwent a multicomponent community-based exercise intervention included in the "More Active Aging Project" (MAAP), which was implemented in 5 municipalities in the West and Ribatejo Regions of Portugal. The study was coordinated by the Faculty of Human Kinetics, University of Lisbon and Sport Sciences School of Rio Maior, between September 2014 and December 2015. All detailed information about the protocol and methods used in the MAAP intervention can be found in Ramalho et al. [27].

A sample of 82 older adults from the abovementioned cohort was recruited through advertising in local centers and sports community services by a multi-stage sample method. Using the results from our previous study [21], the minimum sample size of 61 participants was determined, considering and effect size of $d$ = 0.80, with 80% power and alpha at 0.

To be considered eligible for this study, participants should be 65 years or more, live independently in the community, be autonomous and correctly understand the Portuguese language. Exclusion criteria were considered: self-reported cognitive, neurological, bone and joints, or other impairments that could inhibit the performance of exercises in the standing position autonomously; inability to walk independently and/or without assistance of a walking aid and not having a hip or knee prosthesis.

A written informed consent was obtained from all participants at the beginning of the intervention. The Faculty of Human Kinetics Ethics Committee (Lisbon University) approved the study protocol.

### Exercise program

The MAAP 24-weeks exercise program comprised group-based multicomponent 50-min exercise sessions twice a week. Graduated exercise specialists received 20 hours of training regarding the program methodology and follow-up guidelines during the intervention period. The structure of the exercise program is fully explained elsewhere [27]. In brief, the program aims to develop postural control, balance (static and dynamic), endurance, mobility, walking pattern, and to improve strength and muscle resistance. In order to provide continuous and progressive stimulus to the participants' functional capacities, weekly progressive changes in intensity and complexity, and monthly variation in exercise mode, were incorporated into the program. The progression of the exercise program was controlled through periodic and unscheduled visits by the research team, in order to verify the compliance of the guidelines. Additionally, the instructors recorded a monthly qualitative classification of the participants' performance.

### Assessments

All assessments were conducted at baseline (BL) and after 24 weeks (follow-up, FU) and were administered by examiners who received specific training in applying the test protocols.

To evaluate the eligibility of participants, a health and falls assessment questionnaire designed and validated by the Portuguese Language and Culture [28] was administered through face-to-face interviews. The questionnaire included questions about demographics, health, self-perception status, medication intake, medical history, and falls history. It was used

to verify the eligibility of participants in the study, as well as, identifying health conditions that could prevent participation in the exercise program sessions and interfere with performance in the assessment tests.

All participants were evaluated using the CRS and CS tests on the same day in each phase (BL and FU) by the same examiners. The CS test was chosen as an external reference measure (anchor) in this study, as it measures the same attribute of the CRS (lower limbs strength) and presents results that can be partially explained by the PF strength (β = 0.358, P = 0.074) [29]. The CS test protocol consists of the performance of the maximum possible repetitions of the stand/sit down movements in 30 seconds [19]. The test was administered using an armless chair (height: 43.2 cm), which was supported against a wall to ensure stability. The final score corresponded to the total number of complete cycles performed, i.e., the participants should extend their knees and sit fully on the chair, not lift the feet off the floor and keep the arms folded over the chest. The CRS test protocol is fully described elsewhere [21]. Briefly, the protocol includes the performance of a maximum number of heel lifting / lowering repetitions in the standing position, in 30 seconds, with the knees extended, at maximum possible range and velocity, without transferring the body weight to the hands. The test score corresponded to the number of cycles correctly executed at the end of 30 seconds.

## Data analyses

Descriptive statistical analyses were performed to characterize the sample. Central tendency parameters were determined for continuous variables (mean, standard deviation and median) and relative frequency was calculated for categorical and ordinal variables. The normal distribution of continuous variables was checked with the Kolmogorov-Smirnov Test.

The responsiveness of the CRS test was determined using two different methods: a distribution-based approach and an anchor-based approach.

For the distribution-based approach the results of the two assessment phases were used to calculate the change in scores (FU score—BL score) of the CRS and CS tests. The following statistical parameters, commonly used to assess the responsiveness of instruments, were computed: 1) <u>Effect size</u> (ES)—provides information about the magnitude of change over time by dividing the Mean Change Score of a variable by the SD of its BL Scores[30,31]. To interpret the effect-size data, the cutoff points proposed by Hopkins [32] (ES < 0.20 = trivial effect; 0.20 ≥ ES < 0.60 = small effect; 0.60 ≥ ES < 1.20 = moderate effect; 1.20 ≥ ES < 2.0 = large effect; 2.0 ≥ ES < 4.0 = very large; and ES ≥ 4.0 = nearly perfect); 2) <u>Standardized response mean (SRM)</u>—parameter that indicates if the change of the results over time were large relative to the variability in the measurements. The SRM can be calculated as the Mean Change Score of the variable divided by the SD of the same Change Score. SRM values of 0.20, 0.50, and 0.80 are considered as small, moderate and large change, respectively [33]; 3) <u>Minimal Detectable Change</u> (MDC)—measure that reflects the smallest change in score that can be interpreted as a 'true' change, i.e. beyond measurement error [34]. The formula for the calculation of MDC can be expressed as: 1.96 ($\sqrt{2}$ x Standard Error of Measurement). The calculation of the Standard Error of Measurement was based on the results of our previous study using the following equation: SEM = SD of BL scores ($\sqrt{1-ICC}$). The proportion of participants achieving a degree of improvement beyond the MDC was then determined [35].

The anchor-based approach was performed using a Receiver Operating Characteristic (ROC) curve analysis in order to verify whether the CRS test could discriminate between participants with positive change (improved) versus no change (stable) [36,37]. The cut-off of 3.01 was considered to dichotomize sample in accordance with the minimal detectable change (MDC) determined in a previous test-retest reliability study (CS change score < 3.01 = stable

group; CS change score $\geq$ 3.01 = improved group). An area under the curve (AUC) was used to determine specificity and sensitivity [38], and the cut-off corresponding to the point closer to the upper left corner was defined as the score that best classifies participants who had improved or maintained their state. This cut-off represents the "minimally important difference" (MID) of this test [34], that is, the smallest change in the CRS scores that is considered clinically relevant, or worthwhile to the participants [35], also frequently referred to in the literature as the "minimal clinically important difference" (MCID) [36, 39]. Paired t-tests (or non-parametric Wilcoxon tests) were used to compare data at BL and FU within groups of change.

## Results

Eighty two healthy older adults were eligible to participate in the responsiveness study of the CRS test. All participants were present in both assessment periods and met a minimum attendance threshold in the training sessions. Furthermore, none of the participants showed signs of overexertion, pain in the lower limbs or other signs of discomfort that prevented them from complying with the requirements for a satisfactory assessment.

Participants were mainly women (87.8%) with good general health perception ($\geq$3 points-scale) and a mean BMI of 29.9±5.1 kg/m$^2$ (Table 1), indicating that this group is overweight ($\geq$ 25,0 kg/m$^2$) [40], although out of the range for increased mortality risk (BMI <23.0 or > 33.0) [41].

In general, participants underwent statistically significant improvements in their lower limb strength (CRS and CS, $P < 0.001$). Regarding the distribution-based approach, the effect size was low to moderate, ranging from 0.4 to 0.6 (CRS and CS, respectively) and SRM values were moderate, ranging from 0.5 to 0.7 (Table 2).

The change detected using the CRS was higher, by a statistically significant difference, for the group of participants who improved (CRS change score = 5.8±5.4) when compared with the stable group (CRS change score = 3.0±6.5) (Table 3). Accordingly, it is also possible to verify that the proportion of changes related to initial values in CRS was higher in the improved group (CRS relative change = 37.9 ± 54.9%) than in the stable group (CRS relative change = 15.4 ± 31.5%). The results of the comparison between BL and FU scores validate the

**Table 1. Demographics and functional fitness measures in baseline from the total group of participants and subgroups of CRS scores.**

| Demographic and health parameters | |
|---|---|
| N = 82 | Mean ± SD (median)/ % |
| Age (years) | 72.3 ± 5.0 (72,0) |
| Gender, female (%) | 87.8 |
| BMI (kg/m$^2$) | 29.9 ± 5.2 (29.4) |
| HPS (1–4 scale) | 3,3 ± 0,8 (3,0) |
| **Functional fitness parameters** | |
| N = 82 | Mean ± SD (median)/ % |
| CRS (x/30s) | 25.0 ± 8,8 (24,0) |
| CS (x/30s) | 16,1 ± 4,6 (15,5) |

Data are presented as mean ± standard deviation (median) for continuous variables, and percentage for categorical variables on Baseline.

**Abbreviations**: BMI = Body mass index; HPS = Health Perception Status; CRS = Calf-raise Senior test; CS = 30 s chair stand test.

**Table 2. Responsiveness of FF measures for the Total Sample and by Subgroups of Lower and Higher CRS Scores.**

| Parameters | Change Score (N = 82) | | ES | SRM |
| --- | --- | --- | --- | --- |
| | mean | ± SD (median) | | |
| CRS score (x/30s) | 3.4 | ± 6.6 (3.0) | 0.4 | **0.5***  |
| CS score (x/30s) | 3.6 | ± 5.3 (3.0) | **0.6†** | **0.7***  |

Data are presented as mean ± standard deviation (median). Cohen's Effect Size (ES) and Standardized Response Mean (SRM) from the comparison between baseline and follow up scores. Dagger (†) indicates ES|d| > 0.6 and asterisk (*) indicates SRM > 0.5 (medium to high Effect Size or response mean, here considered as important differences between group means).

**Abbreviations**: CRS = Calf-raise Senior test; CS = 30 s chair stand test; ES: Effect Size; SRM = standardized response mean.

lack of significant changes in the stable group, as expected. Subgroups of change did not reveal equivalence on BL (stables = 27.9± 9.0 vs improved 2.00±6.7; $P$ = 0.006) but the differences were non-significant on FU ($P$ = 0.61).

A ROC curve analysis shows a change score in the CRS test greater than or equal to 3.5 reps has a sensitivity of 0.68 and specificity of 0.33. The area under the curve found in this test was 0.67 ($P$ <0.01) indicating moderate discriminative ability.

Fig 1 shows the proportion of participants in the exercise program who had greater improvements in plantar-flexors (PF) strength and power than the MDC values (42.7%) and beyond MID (47.6%). Thirty five participants reached or overcame the MDC cut-off point of 4.6 and presented a mean CRS change score of 9.60 (± 4.33), while 37 elderly subjects improved PF function (≥ MID of 3.5) and revealed a CRS change score of 9.03 (± 4.45).

## Discussion

This study aimed to test the responsiveness of CRS test, in a sample of older-adult participants undergoing a 24-weeks' community exercise program.

The CRS test showed a small to moderate responsiveness. A higher responsiveness was found for the CS test both in this study, as well as in other studies [42].

The responsiveness of the CRS test was also performed using an external measure to compare changes and establish cut-off values associated with meaningful improvements in physical function as a result of the intervention. The comparative analysis of the CRS scores obtained in the two time points (BL and FU) between the groups of positive change (improved group) and without change (stable group) in the reference test (CS) revealed significant differences in absolute and relative changes. Considering that the attribute assessed in both tests is the same —strength in the lower limbs; and that the two tests are evaluated in a similar way—the number of movements performed in 30s (revealing the same limitations in cognitive and sensory terms); it seems reasonable to state that the CRS test is responsive to discriminate elderly subjects with relevant changes after an exercise intervention program.

The ROC curve analysis revealed an optimal cut-off point of 3.5 repetitions allowing us to establish the minimal important difference for the CRS test. With this analysis it was possible to identify about 70% of the participants who underwent a truly important change after the intervention, while recognizing approximately 30% of the elderly who did not show a real change in their strength in any of the methods. The MDC value revealed that a change score of 4.6 would be required for the resulting change in participant status to be outside the test error range, which is higher than the MID estimate. This is in accordance with other responsiveness studies, in which anchor-based approaches outweigh the values found in the distribution

**Table 3. Responsiveness of CRS test to the 24 weeks-exercise program, considering groups of change in the CS test (stable and improved).**

| Parameters | CS stable group (N = 34) | | CS improved group (N = 48) | | P value | ES (d) |
|---|---|---|---|---|---|---|
| | mean | ± SD (median) | mean | ± SD (median) | | |
| CRS Baseline (x/30s) | 27.8 | ± 9.0 (27,0) | 22.0 | ± 6.7 (23.0) | 0,00** | 1.2† |
| CRS Follow up (x/30s) | 30.8 | ± 9.1 (30,0) | 27.8 | ± 6.5 (28.0) | NS | 0.4 |
| CRS Change score (x/30s) | 3.0 | ± 6,5 (2.0) NS | 5.8 | ± 5.4 (5.0)§ | 0.03* | 0.6† |
| CRS relative change (%) | 15.4 | ± 31.,5 (15.0) | 37.9 | ± 54.,9 (32.0) | 0.02* | 0.5† |

Means ± standard deviations (median) of CRS test results on baseline and follow up; change scores and relative change (%) in groups of participants classified as stable or improved (CS test change scores, MDC = 3.01).

*P<0.05,

**P<0.001, comparison between groups (T-Student or Mann Whitney tests);

§P<0.001, comparison between baseline and follow-up scores (Wilcoxon test);

†ES |d |>0.5 (medium to high Effect Size, here considered as clinically relevant differences between group means); NS indicates non-significant differences.

**Abbreviations**: CRS = Calf-raise Senior test; CS = 30 s chair stand test.

approaches [40,43], highlighting the difficulties in using the former method due to the lack of an optimal threshold which can accurately determine the cut-offs to set the MID. Therefore, if only the value of MID is considered to evaluate the effects of an intervention, the results may indicate that improvements can be attributed to test error and not necessarily to a true change. Moreover, if the MDC is used as the only reference of change, then a small but significant effect can be neglected [40]. Several authors state that the value of the MID may be related to the SEM, depending on the degree of improvement defined by the anchor [44,45]. It has been shown that a cut-off point for MID between "slightly improved" and "moderately improved" may be similar to 2.0–2.3 * SEM [34], while in studies requiring "moderate" or "much improvement", MID corresponds to about 2.5–2.6 times the SEM value [34]. In the present study, the SEM value calculated for the CRS test was 2.8, indicating that the closest cut-off point of this relationship would be the MDC = 2.3 * SEM. In this case, the MID would be on the order of 1.25 * SEM, which is more in line with other studies [44] in which similar values using clinical parameters as the anchor were observed. Therefore, it is suggested that both values should be used in assessing ankle strength improvements resulting from an exercise program. In practice, we can establish that changes in CRS scores below 3.5 must be considered insufficient, values between 3.5 and 4.6 may be viewed as acceptable for slight to moderate changes (but with a chance of being inside the range of measurement errors), while scores above 4.6 can be considered as a true change.

The anchor-based approach used in the present study was based on an ecological perspective, considering that the effects of community programs are usually assessed through field tests, which are easy to administer, have good acceptability and motivation, and reach a large number of participants in screenings [17]. Therefore, taking into account that the CS test is widely used in this context, and that it has been showing very positive indicators of validation, reliability and sensitivity to change in previous studies [19,46,29], its use seems to us acceptable for a preliminary approach in the scope of CRS test responsiveness assessment. Nonetheless, to indicate a true change in the plantar-flexors strength, a comparison with a quantitative direct measures as gold-standard measures would be more accurate, such as a dynamometer strength test, force platform, a biomechanical gait analysis, or the use of other specific clinical test for ankle strength assessment (e.g. manual muscle testing) [47].

The lack of other studies assessing CRS responsiveness prevents comparison of the results found in the present study. As suggested by Revicki [48], the estimate of MID should be

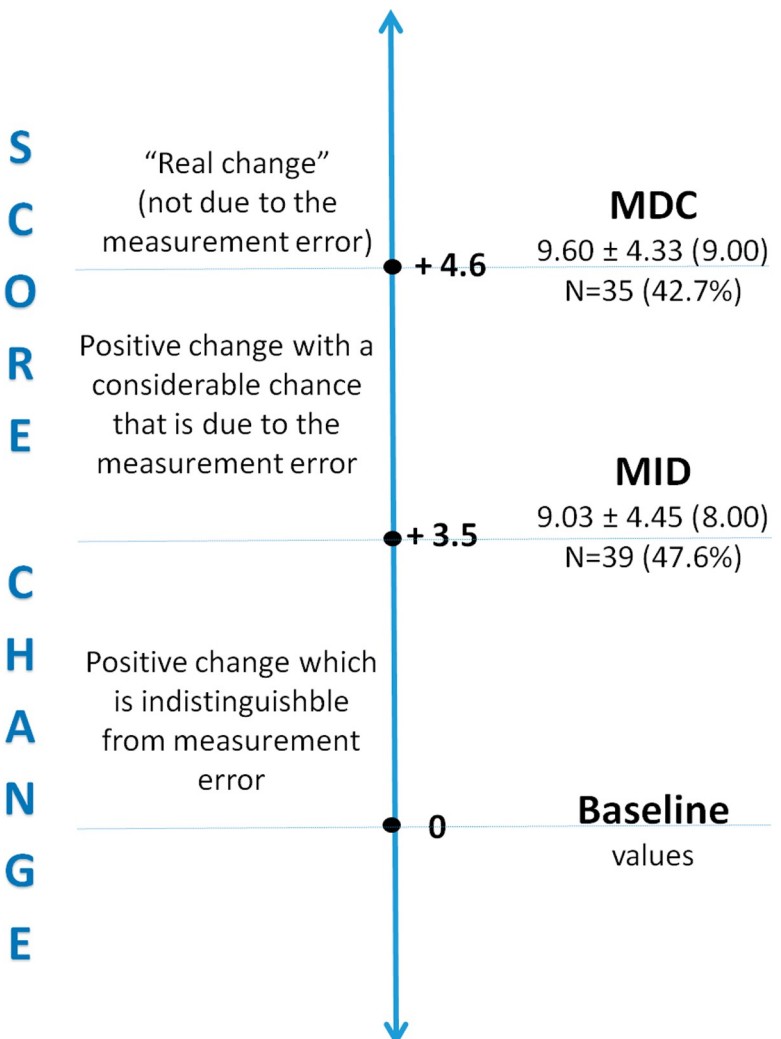

**Fig 1. Means and standard deviations of change scores, and proportion of participants who reached or exceeded the MDC and MID cut-off points.**

confirmed through on the accumulation of evidence from several studies, in order to build greater confidence in the defined cut-offs values. Therefore, it is necessary to develop further studies using a larger sample composed of participants with higher / lower functional-fitness levels, as well as performing interventions that address longer-term changes in physical function. We also suggest carrying out additional research to evaluate whether the test can detect relevant changes in people with a higher baseline physical condition since the sensitivity to change demonstrated by the CRS test was more evident in groups of subjects with lower initial scores. Despite the weaknesses identified, this is the first article that defines the responsiveness of the CRS test, identifying cut-off values of MDC and MID that may help to establish a basis for future studies focused on plantar flexion strength and power interventions in the elderly.

## Conclusions

This study aimed to examine the responsiveness of the Calf-Raise Senior (CRS) test through a 24-week exercise intervention designed to improve muscle strength, endurance, flexibility and balance, as key factors affecting physical function.

The results strengthen the psychometric properties of CRS, revealing its ability to detect change after a 24 week community exercise program focused on improving functional mobility in the elderly. In addition to its excellent validity, reliability, and acceptability by participants and professionals, the CRS test revealed good responsiveness in detecting changes in plantar flexion function over time. The present study also provides data relevant to the field application of these measures, reporting cutoffs of 3.5 and 4.6 for the MDC and MID estimates, respectively.

## Supporting information

**S1 File.**
(ZIP)

## Acknowledgments

The authors are grateful to all of the seniors who voluntarily participated in this study, and acknowledge the valuable assistance of Sara Gabriel and all the other MAAP project technicians who gave their support in data collection.

## Author Contributions

**Conceptualization:** Helô-Isa André, Vera Moniz-Pereira, Fátima Ramalho, Filomena Carnide.

**Formal analysis:** Helô-Isa André, António Veloso, Filomena Carnide.

**Funding acquisition:** Rita Santos-Rocha.

**Investigation:** Helô-Isa André.

**Methodology:** Helô-Isa André, Fátima Ramalho, Filomena Carnide.

**Project administration:** António Veloso.

**Resources:** Rita Santos-Rocha.

**Supervision:** António Veloso, Filomena Carnide.

**Validation:** Filomena Carnide.

**Writing – original draft:** Helô-Isa André, Filomena Carnide.

**Writing – review & editing:** Helô-Isa André, Vera Moniz-Pereira, Fátima Ramalho, Rita Santos-Rocha, António Veloso, Filomena Carnide.

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
