## [Decision Letter · Decision Letter 0]

3 Jan 2020

PONE-D-19-31500

Responsiveness of the Calf-Raise Senior Test in community-dwelling older adults undergoing an exercise intervention program

PLOS ONE

Dear Dr. Andre,

Thank you for submitting your manuscript to PLOS ONE. After careful consideration, we feel that it has merit but does not fully meet PLOS ONE’s publication criteria as it currently stands. Therefore, we invite you to submit a revised version of the manuscript that addresses all the points raised during the review process.

We would appreciate receiving your revised manuscript by Feb 17 2020 11:59PM. To enhance the reproducibility of your results, we recommend that if applicable you deposit your laboratory protocols in protocols.io, where a protocol can be assigned its own identifier (DOI) such that it can be cited independently in the future. For instructions see: http://journals.plos.org/plosone/s/submission-guidelines#loc-laboratory-protocols

We look forward to receiving your revised manuscript.

Kind regards,

Gianluigi Forloni

Academic Editor

PLOS ONE

Journal Requirements:

2. In your Methods section, please provide additional information about the participant recruitment method and the demographic details of your participants. Please ensure you have provided sufficient details to replicate the analyses such as: a) the recruitment date range (month and year), b) a statement as to whether your sample can be considered representative of a larger population, and c) descriptions of where participants were recruited and where the research took place.

Reviewers' comments:

Reviewer's Responses to Questions

**Comments to the Author**

1. Is the manuscript technically sound, and do the data support the conclusions?

Reviewer #1: Yes

2. Has the statistical analysis been performed appropriately and rigorously? 

Reviewer #1: Yes

3. Have the authors made all data underlying the findings in their manuscript fully available?

Reviewer #1: Yes

4. Is the manuscript presented in an intelligible fashion and written in standard English?

Reviewer #1: No

5. Review Comments to the Author

Reviewer #1: Congratulation to the technically sound article.

Please let the article be reviewed by a native speaker. There are to many language errors (especially singular/plural) included making it hard to follow in some passages.

Once this adjustions and the following remarks are handled I'm optimistic, that the article might meet the criterias for publication.

Abstract.:

, which are particularly relevant in mobility -> redundant. remove.

in order to detecting improvements -> grammar

assessed with CRS -> introduce abbreviation in Abstract Line 4 and first occurence in main text and in conclusion

change measures of changes-> change-measures

using -> how? Difference among both scores?

Extended version in Abstract: Effect size (ES, Standardized response mean (SRM, Minimal Detectable Change (MDC

L2: in the last decades -> delete

L25-L26: Among these, few. -> reframe the sentence (few already in the previous sentence)

L28: and “Timed Up and Go” -> and the “Timed Up and Go” test

L31: , which was already been mentioned as having great significance -> , as beeing relevant

L37: then -> delete + the whole sentence must be rewritten

L37: responsiveness it’s not yet been -> awkward

L37: of CRS test, -> of the CRS test,

L51: is described elsewhere [31]. -> A... can be found in ...

Please make all related articles available with the next submission for review purposes. [31]

L83: a health and falls assessment questionnaire designed is there a english speaking representative questionair availebe`? Than please name this as well for orientation

L88: 30-seconds chair stand test (CS) -> unify names among paper based on the naming version in introduction

L89: "The CRS test was developed to evaluate ankle muscle function in older adults and had shown to have an excellent test-retest reliability (ICC = 0.90), inter-rater reliability (ICC = 0.93-0.96) and a good intra-rater agreement (ICC = 0.79-0.84) [24, 25]. It also presented a significant association between their scores and laboratory strength assessments (isometric, r = 0.87, r2 = 0.75; isokinetic, r = 0.86, r2 = 0.74; and rate of force development, r = 0.77, r2 = 0.59), demonstrating to be an excellent indicator of ankle strength in older adults [25]. The CRS test protocol is fully described elsewhere [25]. Briefly, the test includes the performance of a maximum number of heel lifting / lowering repetitions in the standing position, in 30 seconds, with the knees extended, at maximum possible range and velocity, without transferring the body weight to the hands. The test score corresponded to the number of cycles correctly executed at the end of 30 seconds. " -> Move to intro

L100: "The CS test is considered as a good indicator of mobility and has shown to predict the functional decline and risk of falls in the elderly [18, 34]. This test aims at evaluating functional strength of the lower limbs in older adults [33-36] and was chosen as an external reference measure (anchor) in this study since it measures the same attribute of CRS: lower limbs strength. In addition, having an excellent criterion validity, confirmed through a strong correlation with maximum voluntary contraction in the lower limbs (r = 0.77, 95% CI = 0.64-0.85) and an effect size of 0.83 when comparing elderly with high vs. low level of physical activity (P < 0.001), this test has also demonstrated an excellent test-retest reliability (r = 0.89, 95% CI = 0.79-0.93) [19]. The CS protocol consists of the performance of the maximum possible repetitions of the stand/ sits down movements in 30 seconds [37, 38]." -> Move to intro and adjust surrounding texts to fit again

L79: Assessments

Here it might be great to have some extra details regarding the intervention. In this case aklter the Sectiontitle correspondingly

L41: Didn't you included a control group? Just all 61 in one group? Please clarify and state in text. In that case, you can only confirm to measure a change. Its not confirmable that you have a stable base measure, which would b a pitty :)

L122: data at BL and FU within groups. -> groups?

L126ff: The methods in the following Lines must be described super-precise (eventually even formulars if not common known ones)

L130: "These approaches have 131 advantages and limitations, and some authors advise both, in order to provide a most complete 132 and comprehensive information of changes [39, 40, 42]. -> IMO: not relevant for Methods. Ratehr intro in case ...

L133: change scores of -> reframe

L148: important to be considered as “true 149 changes” [49]. -> Please clarify the approach name when a a true change was found. Which conditions must be given in that case, etc...

L151: I might have missed, but where have you explained how to separate these groups?: "between groups of participants with positive change (improved) versus no change (stable) [39, 50].""

L154 / L90: Its challenging that you are using two different ICCs within this study. This must be clearly discussed and reasoned.

L240: The ROC curves analysis revealed an optimal cut-off point of 3.5 repetitions as a change score -> of whoch assessment?

L220: which was lower than -> which was less distinct than

L244: who did not have a real change in their strength. -> who did not have shown a real change in their strength in any of the methods.

L255 : between "slightly improved" and "moderately improved" may be similar to 2.0-2.3 * SEM, -> Cite a corresponding reference

6. PLOS authors have the option to publish the peer review history of their article (what does this mean?). If published, this will include your full peer review and any attached files.

Reviewer #1: No

---

## [Author Response · Author response to Decision Letter 0]

17 Feb 2020

“Congratulation to the technically sound article.

Please let the article be reviewed by a native speaker. There are to many language errors (especially singular/plural) included making it hard to follow in some passages.

Once this adjustions and the following remarks are handled I'm optimistic, that the article might meet the criterias for publication.”

Author response: We found your comments extremely helpful and have revised accordingly. All spelling and grammatical errors pointed out by the reviewer have been corrected in the text.

Abstract:

1) which are particularly relevant in mobility -> redundant. remove.

Author response: Agree. We have modified, accordingly.

2) in order to detecting improvements -> grammar

Author response: Agree. We have modified, accordingly (line 2)

.

3) assessed with CRS -> introduce abbreviation in Abstract Line 4 and first occurence in main text and in conclusion

Author response: We agree with this and have incorporated your suggestion throughout the abstract (lines 5 and 9).

4) change measures of changes-> change-measures

Author response: Agree. We have modified, accordingly (line 10).

5) using -> how? Difference among both scores?

Author response: We have modified the text to make it clearer (lines 10-11).

6) Extended version in Abstract: Effect size (ES, Standardized response mean (SRM, Minimal Detectable Change (MDC

Author response: Agree. We have incorporated your suggestion throughout the abstract (lines 13, 14, 18).

Introduction

1) L2: in the last decades -> delete

Author response: Agree. We have modified, accordingly (line 29)

2) L25-L26: Among these, few. -> reframe the sentence (few already in the previous sentence)

Author response: Thank you for pointing this out. We have modified the text to make the sentence clearer (lines 56-60).

3) L28: and “Timed Up and Go” -> and the “Timed Up and Go” test

Author response: Agree. We have modified, accordingly (line 61)

4) L31: , which was already been mentioned as having great significance -> , as beeing relevant

Author response: Agree. We have modified, accordingly (line 65)

5) L37: then -> delete + the whole sentence must be rewritten

Author response: We have modified the sentence to make it clearer (lines 75-81).

6) L37: responsiveness it’s not yet been -> awkward

7) L37: of CRS test, -> of the CRS test

Author response: Agree. We have modified, accordingly (line 74-76)

Materials and Methods

8) L51: is described elsewhere [31]. -> A... can be found in ...

Please make all related articles available with the next submission for review purposes. [31]

Author response: We have modified, as requested (lines 95-96). We also put the reference’s DOI in the bibliography section so that they can be consulted, if necessary.

9) L83: a health and falls assessment questionnaire designed is there a english speaking representative questionair availebe`? Than please name this as well for orientation

Author response: Thank you for your question. Unfortunately the English version of the QESQ questionnaire was not yet published as a journal article, but only as an abstract in a journal supplement (congress proceedings). It can be consulted trough the DOI:10.1249/01.mss.0000477095.65735.9c

10) L88: 30-seconds chair stand test (CS) -> unify names among paper based on the naming version in introduction

Author response: Agree. We have incorporated your suggestion throughout the article

11) L89: "The CRS test was developed to evaluate ankle muscle function in older adults and had shown to have an excellent test-retest reliability (ICC = 0.90), inter-rater reliability (ICC = 0.93-0.96) and a good intra-rater agreement (ICC = 0.79-0.84) [24, 25]. It also presented a significant association between their scores and laboratory strength assessments (isometric, r = 0.87, r2 = 0.75; isokinetic, r = 0.86, r2 = 0.74; and rate of force development, r = 0.77, r2 = 0.59), demonstrating to be an excellent indicator of ankle strength in older adults [25]. The CRS test protocol is fully described elsewhere [25]. Briefly, the test includes the performance of a maximum number of heel lifting / lowering repetitions in the standing position, in 30 seconds, with the knees extended, at maximum possible range and velocity, without transferring the body weight to the hands. The test score corresponded to the number of cycles correctly executed at the end of 30 seconds. " -> Move to intro

Author response: Thank you for pointing this out. We moved the first part of this sentence to the introduction section and have modified the surrounding text to make it clearer (lines 68-73). The second part of the sentence (underlined) was modified and maintained in this section in order to describe the CRS protocol together with the CS protocol for the readers (lines 155-160).

12) L100: "The CS test is considered as a good indicator of mobility and has shown to predict the functional decline and risk of falls in the elderly [18, 34]. This test aims at evaluating functional strength of the lower limbs in older adults [33-36] and was chosen as an external reference measure (anchor) in this study since it measures the same attribute of CRS: lower limbs strength. In addition, having an excellent criterion validity, confirmed through a strong correlation with maximum voluntary contraction in the lower limbs (r = 0.77, 95% CI = 0.64-0.85) and an effect size of 0.83 when comparing elderly with high vs. low level of physical activity (P < 0.001), this test has also demonstrated an excellent test-retest reliability (r = 0.89, 95% CI = 0.79-0.93) [19]. The CS protocol consists of the performance of the maximum possible repetitions of the stand/ sits down movements in 30 seconds [37, 38]." -> Move to intro and adjust surrounding texts to fit again

Author response: Thank you for this suggestion. We removed completely the first part of the sentence because we considered that it was cutting the flow of the text and did not add information that was relevant to the article. The second part of the sentence (underlined) was modified and maintained in this section in order to describe the CS protocol for readers (lines 150-155).

13) L79: Assessments. Here it might be great to have some extra details regarding the intervention. In this case aklter the Sectiontitle correspondingly

Author response: Thank you for pointing this out. Although we agree that this is an important consideration, we think that it not appropriate for inclusion in this manuscript because all information related to the intervention was described in detail in the referenced study, since it is a specific publication about the study protocol. We believe that the information provided in this manuscript can be sufficient to understand the results of the study. However, if you still feel that it is necessary to include more information about the intervention in this manuscript, we are available to change, accordingly.

14) L41: Didn't you included a control group? Just all 61 in one group? Please clarify and state in text. In that case, you can only confirm to measure a change. Its not confirmable that you have a stable base measure, which would b a pitty :)

Author response: You have raised an important aspect that can be pointed out as a potential limitation of the study, since the inexistence of control group did not allow the use of the Guyat’s index of responsiveness measure (Guyat G., 1987), which could strengthen the study. However, and considering the methodology of responsiveness analysis used, it is possible to state that this study was in line with most studies in this area, where the one-group design sample is usual. Responsiveness was demonstrated in this study both by distribution-based and anchor-based methods, which can be considered a strong point, as it allowed to confirm the test's responsiveness through different routes.

With regard to distribution-based analysis, according to authors such as Strand (2011) and Terwee (2007), the ability of an instrument to respond to change is mainly related to the minimally detectable change (MDC). This measure reflects the smallest change in a score that can be interpreted as a real change above the measurement error, and is dependent on the test-retest reliability of the instrument. In this case, we used the MDC values defined for the CRS and CS in previous test-retest reliability studies to verify whether the change score of the participants was above this measure, as an indication of real change.

Regarding the Anchor-Based Approach, we used the dichotomized scores of the CS as an external anchor of meaningful change. As referred by Strand (2011), ability of changes in scores on the physical tests to distinguish between participants who had improved, and those who did not have improved, can be analyzed using external standards of meaningful change by the receiver operating characteristic (ROC) curve. Thus, the stable measure in this study was considered as the group of participants who did not have improved their scores in CS test (anchor), considering the value of 3.01 for the change score.

We can naturally point out that this study would have achieved more robust results if a gold standard measure was used as an anchor, as mentioned in the limitations section. Likewise, the use of self-report questionnaires of functioning to identify the participants' perception of individual change in this parameter could add more information about the test's responsiveness. 

Even so, we believe that our study complies with what would be necessary to affirm that the results found can be valuable to support the use of the CRS test in interventions with exercise for the elderly.

15) L122: data at BL and FU within groups. -> groups?

Author response: Thank you so much for catching this confusing sentence, which we have now been corrected. The sentence was moved from lines 166-169 to the end of the section (lines 228-229) because we realized that the methodology used to dichotomize the sample (improved vs. stable group) had not yet been explained at the beginning of the section, which could be confusing for the reader. We also removed information about correlations and comparisons between groups, as they were cutting the flow of text and did not add information that could be relevant to the article.

16) L126ff: The methods in the following Lines must be described super-precise (eventually even formulars if not common known ones)

Author response: We agree with this and changed the text to make it clearer, removing irrelevant parts and adding some extra information to facilitate the reader's understanding (lines 170-204).

17) L130: "These approaches have 131 advantages and limitations, and some authors advise both, in order to provide a most complete 132 and comprehensive information of changes [39, 40, 42]. -> IMO: not relevant for Methods. Ratehr intro in case ...

Author response: Thank you for this suggestion. We agreed that the text was not adding relevant information to the article and we removed the sentence completely. 

18) L133: change scores of -> reframe

Author response: We agree. The text was completely changed in this part, as you suggested (lines 180-183).

19) L148: important to be considered as “true 149 changes” [49]. -> Please clarify the approach name when a a true change was found. Which conditions must be given in that case, etc...

Author response: Thank you for pointing this out. Extra information was added in the line 197 to clarify the sentence.

20) L151: I might have missed, but where have you explained how to separate these groups?: "between groups of participants with positive change (improved) versus no change (stable) [39, 50].""

Author response: We agree. The text was changed by including more information to provide a better comprehension about the methodology used (212-216).

21) L154 / L90: Its challenging that you are using two different ICCs within this study. This must be clearly discussed and reasoned.

Author response: Thank you for your suggestion. We changed the text accordingly, removing the ICC related to the previous study, so as not to confuse readers and facilitate their understanding (line 213).

22) L220: which was lower than -> which was less distinct than

Author response: We agreed that the sentence was not clear. Thanks for your suggestion, but we changed the text differently to facilitate the reader's understanding (lines 291-292).

23) L240: The ROC curves analysis revealed an optimal cut-off point of 3.5 repetitions as a change score -> of whoch assessment?

Author response: Thank you for your question. We agreed that the sentence was not clear and we changed the text provide a better comprehension of the method (lines 313-316)

24) L244: who did not have a real change in their strength. -> who did not have shown a real change in their strength in any of the methods.

Author response: Thank you for your suggestion. We changed the text accordingly (line 319)

25) L255 : between "slightly improved" and "moderately improved" may be similar to 2.0-2.3 * SEM, -> Cite a corresponding reference

Author response: Thank you for pointing this out. The reference was added to the line 332.

Additional clarifications 

In addition to the above comments, all spelling and grammatical errors pointed out by the reviewers have been corrected. 

We also introduced other changes to the article, which aimed to facilitate its understanding and prevent the reader's attention from being diverted. In this way, some parts of the text were removed, or had their structure changed, such as: 

• Tables - were altered so that the data was displayed with only one decimal place, since in the previous format (two decimal places) it hindered the observation of the data;

• Lines 88-90 – sentence removed. We considered that it did not add relevant information to the manuscript;

• Lines 171-178 – most part of the text were removed. The feedback provided by the reviewer about this section allowed us to conclude that this part of the text was excessive and could lead the reader to not retain the desired information about the methodology used; 

• Lines 303 to 312 – the first paragraph was moved to the lines 350-353 since we believed that it would be better framed as a limitation of the study. The second paragraph, in turn, we consider that did not make a great contribution to the discussion of the results, and in this way, it was eliminated.

---

## [Editor Report · Decision Letter 1]

26 Mar 2020

Responsiveness of the Calf-Raise Senior Test in community-dwelling older adults undergoing an exercise intervention program

PONE-D-19-31500R1

Dear Dr. Andre,

We are pleased to inform you that your manuscript has been judged scientifically suitable for publication and will be formally accepted for publication once it complies with all outstanding technical requirements.

With kind regards,

Gianluigi Forloni

Academic Editor

PLOS ONE
---

## [Editor Report · Acceptance letter]

6 Apr 2020

PONE-D-19-31500R1 

Responsiveness of the Calf-Raise Senior Test in community-dwelling older adults undergoing an exercise intervention program 

Dear Dr. André:

I am pleased to inform you that your manuscript has been deemed suitable for publication in PLOS ONE. Congratulations! Your manuscript is now with our production department. 

With kind regards,

on behalf of

Dr. Gianluigi Forloni 

Academic Editor

PLOS ONE